# PRUNE AND TUNE: IMPROVING EFFICIENT PRUNING TECHNIQUES FOR MASSIVE LANGUAGE MODELS

**Aaquib Syed, Phillip Huang Guo, & Vijaykaarti Sundarapandiyan**[*]
University of Maryland
College Park, Maryland 20742, USA
`{asyed04, phguo, vsundar1}@umd.edu`

## ABSTRACT

Massive language models with billions of parameters have significant compute expenses and thus can benefit from pruning. Pruning techniques for massive models are typically iterative and require extensive weight retraining after pruning. SparseGPT, a recently introduced one-shot technique for pruning such models, enables pruning without retraining. We improve upon SparseGPT by fine-tuning during pruning with minimal training steps, and we perform experiments against magnitude pruning and find that our iteratively fine-tuned SparseGPT models significantly outperform their magnitude pruning counterparts at high sparsity.

## 1 INTRODUCTION

Large Language Models (LLMs) have become immensely popular in natural language processing due to their unprecedented performance on a variety of tasks. However, state-of-the-art Generative Pre-trained Transformer models use billions of parameters and continue to scale up their parameter count for even better performance (Wei et al., 2022), requiring significant energy and compute for training and inference (Bender et al., 2021). Thus, the previous literature has proposed techniques to prune LLMs while preserving accuracy (Guo et al., 2019; Wang et al., 2020).

In the literature, pruning techniques are largely classified as either iterative or one-shot (Zhang et al., 2022a). Iterative pruning techniques use some heuristic to prune weights of a model, then retrain the remaining weights of the model to regain accuracy, then re-prune/retrain again. For large models, this latter step can be quite expensive as the model may have to undergo extensive retraining with as many steps as original training took (Samar, 2022).

Recently, Frantar & Alistarh (2023) proposed SparseGPT, a one-shot pruning method that works for massive language models that updates non-pruned weights without retraining in order to maintain accuracy. The original paper proposes follow-up research on fine-tuning SparseGPT models to further improve model performance on higher sparsities. We follow up on the original SparseGPT research to examine fine-tuning models after pruning. Additionally, we propose an iterative pruning technique that prunes models to a higher degree of sparsity while maintaining performance.

## 2 METHODS

We implement the SparseGPT pruning method along with magnitude pruning (pruning the lowest magnitude weights) for LLMs. We selected magnitude pruning as a benchmark to compare SparseGPT with as Samar (2022) demonstrated that iterative magnitude pruning with fine-tuning can succeed for multi-billion parameter models. Due to limited compute, we prune 125 million and 1.3 billion parameter models in the open-source Meta OPT family (Zhang et al., 2022b).

We test SparseGPT across three areas: pruning alone, pruning and fine-tuning once (i.e., non-iterative), and iterative pruning and fine-tuning. We evaluate all of our models using perplexity, a standard information-theoretic assessment of model predictive power (Chen et al., 1998), on the WikiText2 test corpus and fine-tune our models on the WikiText2 train corpus. We limit model

---

[*]All authors provided equal contributions.

fine-tuning to only 1000 training steps (see A.1.2) since fully retraining the model would defeat the one-shot purpose of SparseGPT. By keeping the number of training steps small, we maintain SparseGPT's computational advantages while being able to compare relative performance.

## 3 EXPERIMENTS

We conduct experiments to examine fine-tuning SparseGPT. In total, we test SparseGPT one-shot pruning, non-iterative SparseGPT pruning and fine-tuning, iterative SparseGPT pruning and fine-tuning, magnitude one-shot pruning, and iterative magnitude pruning and fine-tuning.

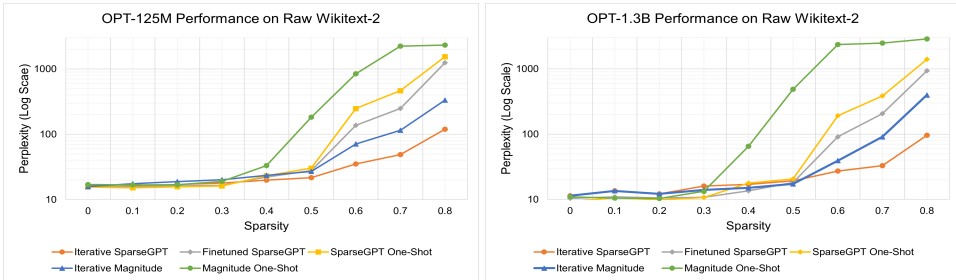

Figure 1: The log-scale graphs of the pruned model perplexity scores on Wikitext-2-raw-v1. "Fine-tuned" denotes non-iterative pruning then fine-tuning. Left: OPT-125M, right: OPT-1.3B.

As the graphs in Figure 1 demonstrate, SparseGPT iterative pruning and fine-tuning is the best technique beyond 0.4 sparseness on OPT-125M and 0.6 sparseness on OPT-1.3B. We find that SparseGPT non-iterative pruning and fine-tuning is better than no fine-tuning in all cases, but is beaten out significantly by both iterative pruning and fine-tuning methods beyond 0.5 sparseness.

Comparing like techniques across both 125M and 1.3B, we find that SparseGPT iterative and magnitude iterative pruning perform similarly up to .5 sparseness, and then SparseGPT performs exponentially better than magnitude beyond .5 sparseness. We also replicate the results in Frantar & Alistarh (2023) by showing that SparseGPT one-shot pruning is better in all cases than magnitude one-shot pruning. The exact values of the models at each sparsity and linear-scale graphs can be seen in A.2.

## 4 DISCUSSION

Our key result is showing iteratively fine-tuning SparseGPT models, even with limited steps, enables better relative model performance at higher sparsity. Additionally, iterative fine-tuning allows further pruning of models up to 80% sparsity with limited perplexity loss. We hope these encouraging results can help provide better accessibility of LLMs to those without the necessary resources to load and infer on the dense models. Furthermore, we hope our work can aid in reducing the carbon footprints of these models by reducing the energy costs of inference.

As an additional impact, magnitude pruning has been shown to result in "lottery ticket" models which have the ability to be pruned and then retrained to similar or higher levels of accuracy than the original (Frankle & Carbin, 2019). Our results suggest that SparseGPT may serve as a more effective strategy for finding better lottery tickets. Furthermore, since SparseGPT-pruned models don't require as much retraining, we hypothesize that lottery tickets can be searched for and discovered faster with SparseGPT, aiding in the overall discovery of better lottery tickets.

### 4.1 FUTURE WORK

As we only examined smaller models, future research pruning models that are closer to the state-of-the-art in both parameter size and training regime is needed. Additionally, testing the lottery ticket hypothesis with SparseGPT pruning and fine-tuning as well as testing alternative pruning schedules for iterative fine-tuning (see A.1.1) that match the recent literature would be beneficial.

URM STATEMENT

The authors acknowledge that all authors of this work meet the URM criteria of ICLR 2023 Tiny Papers Track.

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

# A  APPENDIX

## A.1  FURTHER METHODS

### A.1.1  ITERATIVE PRUNING AND FINE-TUNING

We modify the SparseGPT pruning method to allow for iterative pruning: in every iteration, we force the mask of zeros to include all previously pruned weights so that they will not be overridden. The magnitude pruning method by default does this since pruned weights have the lowest possible magnitude of 0. We test iterative pruning and fine-tuning using a pruning schedule of sparsities. For example, with a pruning schedule of $[.2, .4, .8]$, the model is first pruned to 20% sparsity and fine-tuned (with all of pruned weights staying 0), then pruned to 40% overall sparsity (with the original 20% of pruned weights staying pruned) and fine-tuned, then finally to 80% sparsity and fine-tuned. For all of the iterative models in our experiments, we use a pruning schedule of sparsities between 0 and .8 in increments of 0.1.

### A.1.2  TRAINING HYPERPARAMETERS

For every fine-tuning process, we use the AdamW optimizer with a learning rate of 1e-5. We train on 1000 steps with only 1 epoch for all our experiments. This way, both pruning techniques are given the same number of fine-tuning steps for each sparsity level to make our comparisons as fair as possible. Note that overall, iterative pruning and fine-tuning technically use more training steps for higher sparsities than non-iterative pruning and fine-tuning since they undergo multiple fine-tuning loops.

### A.1.3  REPRODUCIBILITY

We provide our entire codebase for reproducibility (GitHub link). The codebase has the functionality to mask models using both SparseGPT pruning and magnitude pruning, fine-tune models without affecting the pruned weights, and generate graphs of the model performances as shown in the paper.

## A.2  TABLE OF PERPLEXITY SCORES

It is hard to compare the relative performance of the pruned models at low levels of sparsity using just Figure 1. Thus, we provide the exact perplexity scores of the pruned models.

Table 1: OPT-125M performance (perplexity) on raw Wikitext-2

| Sparsity | Iter. SGPT | FT SGPT | SGPT 1-Shot | Mag. 1-Shot | Iter. Mag |
|----------|-----------|---------|-------------|-------------|-----------|
| 0.0 | **15.5491** | 16.4101 | 15.5491 | 17.0491 | 15.8655 |
| 0.1 | 16.0961 | 15.7595 | **15.1254** | 16.7757 | 17.5614 |
| 0.2 | 17.0278 | 16.1470 | **15.6511** | 16.9124 | 18.8294 |
| 0.3 | 17.9596 | 16.5344 | **16.1768** | 18.9990 | 20.0973 |
| 0.4 | **19.8166** | 22.1526 | 23.2115 | 33.2227 | 23.5373 |
| 0.5 | **21.6736** | 27.7708 | 30.2461 | 183.0424 | 26.9772 |
| 0.6 | **35.2026** | 137.5923 | 247.3005 | 842.4491 | 71.2146 |
| 0.7 | **48.7316** | 247.4137 | 464.3548 | 2236.3359 | 115.4521 |
| 0.8 | **119.1989** | 1245.6981 | 1542.4082 | 2314.5171 | 333.2517 |

Table 2: OPT-1.3B performance (perplexity) on raw Wikitext-2

| Sparsity | Iter. SGPT | FT SGPT | SGPT 1-Shot | Mag. 1-Shot | Iter. Mag |
|---|---|---|---|---|---|
| 0.0 | 11.3793 | 10.3865 | **9.1898** | 10.9398 | 11.3793 |
| 0.1 | 13.5108 | 10.9680 | 10.6489 | **10.4941** | 13.5108 |
| 0.2 | 12.1721 | 10.5371 | **9.8145** | 10.3464 | 12.1721 |
| 0.3 | 16.0246 | 10.7993 | **10.7234** | 13.3030 | 14.0246 |
| 0.4 | 17.0493 | **13.5449** | 17.7923 | 65.2450 | 15.0493 |
| 0.5 | 19.2984 | 18.1016 | 20.5491 | 486.0790 | **17.2984** |
| 0.6 | **27.2359** | 90.9736 | 190.9837 | 2351.1123 | 39.2359 |
| 0.7 | **32.8878** | 205.8477 | 385.1699 | 2487.8091 | 90.8878 |
| 0.8 | **96.0358** | 933.2617 | 1398.8472 | 2876.3818 | 397.0358 |

## A.3 LINEAR-SCALE GRAPHS OF PERPLEXITY SCORES

We also provide the linear-scale graphs of perplexity for our two models to emphasize the difference in model performance at high sparsity. We used the log-scale graphs in the paper because model performances at low sparsities are indiscernible in the linear-scale graph.

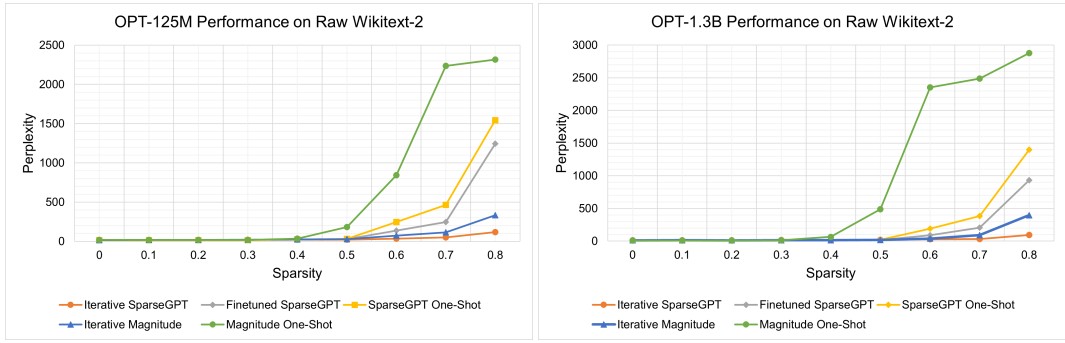

Figure 2: The linear-scale graphs of the pruned model perplexity scores on Wikitext-2-raw-v1. Left: Pruned versions of OPT-125M. Right: Pruned versions of OPT-1.3B.

