# OpenReview forum: "Prune and Tune: Improving Efficient Pruning Techniques for Massive Language Models"
_ICLR.cc/2023/TinyPapers — Submitted to Tiny Papers @ ICLR 2023_

### Official Review · Reviewer_aZDA · 2023-03-23

**Confidence:** 4

**Summary Of Contributions:**

Summary: The authors experiment with augmenting a previously established one-shot pruning method for large language models, SparseGPT, with a fine-tuning step. They find that doing so improves the tuned models' accuracy as measured by perplexity scores on the WikiText2 corpus.

**Rating:**

High Potential (HP): a submission which meets the reviewing criteria and has potential to make an impact on the field

**Strengths And Weaknesses:**

Strengths:
- The fine-tuning idea proposed by the authors is intuitive and worth exploring.

Weaknesses:
- Would need extensively more validation in terms of datasets and varying number of retrain steps to be publication ready.


**Suggested Changes:**

- Define perplexity.
- Be clearer about what fine-tuning is, and give a sense for when fine-tuning becomes a complete retrain.

---

### Meta-Review · Area_Chair_QPRc · 2023-04-08

**Recommendation:** Invite to present (notable)
**Confidence:** 4

**Metareview:**

This paper explores pruning techniques for language models, building on the one-shot pruning method SparseGPT. The authors focus on the effect of fine-tuning after applying SparseGPT, and in particular consider either

- pruning with SparseGPT once, then fine-tuning for a small number of iterations
- iteratively pruning with SparseGPT, fine-tuning, then pruning again and fine-tuning again, repeated over a number of iterations and increasing sparsity levels.

The authors evaluate these approaches by pruning Meta OPT models and fine-tuning them on the WikiText2 corpus, and demonstrate that iterative pruning and fine-tuning with SparseGPT performs very well relative to the other approaches, in addition to replicating findings from previous work. They also provide many details on their experimental results and promise to release code, which seems very useful for reproducing these findings.

Reviewer aZDA found the idea "intuitive and worth exploring" and noted the high potential for impact, although they also noted that a few terms could be clarified and that experimental validation could be more thorough.

Overall I believe this work definitely meets the criteria for Clarity, Correctness, and Reproducibility. The results also seem potentially impactful for enabling more accessible LLM research, and I was impressed by the detail given about the empirical results, so I recommend this work be invited to present as a notable submission.

**Summary:**

The authors evaluate iterative fine-tuning methods for LLMs, building on SparseGPT. The ideas are intuitive, well-explained, and have a high potential for impact, and the work meets the CCR criteria.

**Comments And Feedback To The Authors:**

I would suggest that the authors use consistent notation for the methods being compared in Section 3 and Figure 1. It was not immediately clear to me that "Iterative SpareGPT" was also finetuned during the iterative process, or that "Finetuned SparseGPT" specifically referred to the non-iterative version.

I noticed that the appendix states that iterative fine-tuning uses more training steps due to having multiple fine-tuning loops. Have you considered evenly dividing the total amount of fine-tuning to more fairly compare with the one-shot finetuned method? For instance, if the iterative approach does 3 fine-tuning loops, perhaps you could try doing one-shot pruning followed by 3 epochs of fine-tuning, so that the total number of fine-tuning steps is held constant.

**Reason For Not Giving A Higher Recommendation:**

N/A

**Reason For Not Giving A Lower Recommendation:**

Reviewer aZDA notes that additional empirical validation would improve the results. I agree this would be an improvement, but I believe the work meets the CCR criterion and is a notable submission as it is. (Perhaps the authors can consider this for an expanded version of this submission.)

A few things could be clarified (e.g. the definition of "perplexity" and more details on how much fine-tuning is performed, as noted by Reviewer aZDA) but these seem fairly minor and can hopefully be fixed easily in the next revision.

---

### Decision · Program_Chairs · 2023-04-09

Invite to present (notable)